# Examination of the Effect of Career Barriers and Presenteeism Behaviors on Teacher Professionalism through Structural Equation Model

**Nuri Berk Güngör** [1], **Serkan Kurtipek** [2], **Fikret Soyer** [1,*], **Ekrem Levent İlhan** [2]
**and Laurentiu-Gabriel Talaghir** [3,4,*]

1   Sports Sciences Faculty, Balıkesir University, 10145 Balıkesir, Turkey
2   Sports Sciences Faculty, Gazi University, 06560 Ankara, Turkey
3   Faculty of Physical Education and Sport, Dunarea de Jos University of Galati, 80000 Galați, Romania
4   Department of Physical Education and Health, South Ural State University, 454080 Chelyabinsk, Russia
*   Correspondence: fikretsoyer@gmail.com (F.S.); gtalaghir@ugal.ro or gtalaghir@gmail.com (L.-G.T.)

**Abstract:** The aim of this study is to reveal the effect of career barriers and presenteeism behaviors of physical education and sports teachers on teacher professionalism through a structural equation model. The hypotheses presented because of the literature review were tested with the fictional model. The sample of the study consists of a total of 411 physical education and sports teachers, who work in Ankara. As data collection tools, the Teachers' Career Barriers Scale, the Stanford Presenteeism Scale and the Teacher Professionalism Scale were used in the research. In the analysis of the data, the theoretically created model was tested through a structural equation model. Considering the findings obtained from the research, it can be stated that the participants are faced with career barriers, that they exhibit their presenteeism behaviors at a reasonable level, and that the professionalism level is above average. On the other hand, it was concluded that male participants encountered more career barriers and displayed more presenteeism behavior, while female participants had a higher professionalism level than that of male participants. In addition, it was determined that administrative barriers affect teacher professionalism, while presenteeism behaviors do not have an impact on professionalism level.

**Keywords:** career barrier; presenteeism; professionalism; teacher

## 1. Introduction

The term 'career' is the name given to the patterns and processes of professions and positions that people occupy throughout their working lives [1]. Career includes the whole of activities related to self-development and promotion of the individual during his/her working life [2,3]. A career consists of the knowledge and experience acquired during one's working life, and it enables him/her to specialize in the relevant branch as s/he develops [4]. The concept of career is not only about people with high status or with chances of rapid progress, it is a development process that is acquired in different ways [5].

Although career is acknowledged by most individuals as simply choosing a profession and progressing in it, it is actually a life-structuring process with ups and downs that affect all areas of the life of the individual [6]. Therefore, it covers the entire working life of individuals. The career of an individual is a process that begins with the graduation from school and entry into the business life and continues through vertical or horizontal mobility within the organization, and ends with retirement [3]. In this process, the tendency of individuals to advance in their business life has always existed and will continue to exist. This tendency stems from the nature of human beings [7]. Organizations are one of the most fundamental structural elements that have the potential to affect the career of the individual in the working climate.

There may be events or conditions that can complicate career development, internally or externally, which is a process that requires mutual interaction between organizations and employees [8]. The factor that employees consider to be a barrier to their promotion in working life is called career barriers [9]. Individuals' personal thoughts and initiatives related to career can also be seen as a barrier [10]. Career barriers can originate from the individual, the work environment, or a combination of the two. Loss (e.g. in a person's job or support system), handicap (reduced physical or mental ability through aging, injury or chronic illness), change (a new job, employer, workplace), conflicts (between people or roles), increased expectations or job demands, discrimination (unfair treatment/harassment), underemployment (not using someone's skills), negative performance feedback and negative aspects of positive events (e.g. promotion responsibility, demands and related stress) can be among the barriers [11].

Career barriers explain how and why people engage with the organization. The employee does not want to get a job in a place where the need for a career barrier is not met [12]. Therefore, it is seen that as career barriers increase, various negative outcomes emerge [13]. It is an inevitable fact that the experienced career barriers will cause some consequences that will decrease the productivity of the employees, and accordingly, the decrease in employee productivity will reflect on the productivity of the organization. At this point, the important thing is to determine the effective methods of dealing with perceived barriers and such barriers [14]. It is considered that barriers are not insurmountable and can be overcome, although they have varying degrees of difficulty depending on the particular barrier and individual [8]. In this context, another noteworthy concept about the negative situations encountered in organizations is presenteeism. Presenteeism behavior, which is among the behaviors that individuals can exhibit in their work environments, has recently become an important concept to be examined in terms of businesses [15].

Presenteeism behavior refers to the employees' behaviors of engaging in personal activities rather than work-related activities while at work [16]. Although presenteeism is defined as the presence of the employee in the workplace in a sick state, it is possible to see from some of the behaviors that the employee exhibits in the workplace. The fact that the employee does some of the things that they has to do outside of their duties in the workplace within working hours gives the signals of the presenteeism behavior. Some sample behaviors related to presenteeism are mailing with friends or family members, spending time (surfing) on the internet instead of working, making personal payments (online transactions), arranging appointments such as doctor, hairdresser, etc. in the course of working hours, watching television, playing computer games and online shopping [17,18].

Although the existence of presenteeism does not seem very important at first, it is important for organizations in the long term and causes large amounts of financial loss [19,20]. Therefore, presenteeism can cause serious problems in the physical and mental health of employees in the long term [21]. The low productivity problem arose because the employees were working when they were not healthy, and businesses had to bear high hidden costs due to this tendency [22]. Therefore, the effect of pretending to work in the efficiency and performance indicators of organizations (presenteeism) is quite high [23–25].

From the employee's point of view, presenteeism is important as it worsens existing medical conditions, harms the quality of working life, and causes impressions of ineffectiveness in the workplace due to reduced productivity [26]. In addition to the career development of the individual, it is considered that the presenteeism can negatively affect the basic purpose of existence and expertise in the profession, that is, professionalism.

Every profession in various fields has values that should be used as a guide in relevant professions [27]. Instead of negative behaviors that do not overlap with these values, it is important to reinforce positive behaviors that will increase the total quality. The concept of professionalism, which is one of the positive behavior patterns in professional life, comes to the fore in the professional life. Professionalism, which can be defined as specializing in

the job and maintaining the career development by having knowledge and skills, stands out as one of the important factors affecting the productivity of employees.

Today, the concept of professionalism is widely used all over the world, but there is no clear definition of what exactly it is. It is thought that this is due to different meanings attributed to the concept of profession and the origin of professionalism [28]. Professionalism refers to the commitment of members of the profession to developing their professional skills and to continuously improving the strategies they use while performing tasks appropriate to their profession [29]. Everyone who embraces career development wants to be recognized as a professional in their field [30]. Each profession has developed unique ethical codes and professional behavior patterns. It can be said that owing to these behavioral patterns, people know their professional boundaries better and they do not have difficulty in adapting to these limits thanks to the standards established [31]. According to one view, the professional can be characterized by using four basic elements: basic knowledge, promise of competence, financial resources, and educational conditions [32]. However, there are different approaches that describe professionalism and highlight its different features.

Professionalism is a dynamic and demanding process. Whether an individual is a professional or not can be evaluated in terms of fulfilling some criteria for their field of job and can be defined in the context of fulfilling these criteria [33]. Although different principles of professionalism have been adopted in every field, professionalism is based on important principles in the field of education and training. As a matter of fact, the quality of the teacher is an important factor that will directly affect student development. Teacher professionalism implies that the inherent responsibility of teachers can be realized to develop expertise and dedication in the world of education and can be applied scientifically as well as in their professional fields. In other words, teachers should be able to develop various competencies unique to them across sectors [34]. Positive career attitudes such as professionalism are an inseparable part of teachers, as they will then instill them in their students [35].

When many regulations are examined, it is seen that teachers should have four competencies in general: pedagogical competence, professional competence, social competence and personality competence [36] It is stated that the performance and professionalism of a teacher is equivalent to the ratio of fulfilling the elements such as specialization and development in teaching material, high commitment to the job, discipline in teaching, creativity in teaching and cooperation [37,38].

Teachers who embrace professionalism are teachers who want to prioritize the quality of services and products. Teacher services should meet the standards of needs of society, nation and users, and should maximize students' abilities by arranging them in line with each student's potential and skills [39]. In addition to the depth and breadth of teachers' pedagogical aspects, the reflection of the teacher in the learning process is also an important aspect of teacher professionalism [38]. Therefore, the quality of the teacher can be understood from their professionalism [40]. Professional teachers are always considered to be people who constantly improve their competencies, are always creative, innovative and analyze the strengths and weaknesses of what is done in the teaching and learning process. In addition, individuals can be affected by ordinary and extraordinary environmental factors and sociological changes.

A different aspect of this research is that it is carried out during the COVID-19 Pandemic process. Thus, although this process primarily affects health and economy, it is predicted that teachers may also be affected by the rapid changes in educational practices, and that this may directly reflect on the quality of education.

It is essential to have teachers who face many difficulties in the effort to improve the quality of education, and who are qualified to realize the professional, modern nuances of education through adequate welfare support and maintaining legal certainty. It can be stated that it is important for teachers to exhibit professional behaviors, which show that they are experts in their profession in achieving these goals [41]. In this context, it is

important to determine the career barriers, presenteeism behaviors and professionalism characteristics of physical education and sports teachers. The research was carried out in order to examine the interaction between these three phenomena and suggestions were made as a result of the research.

## 2. Materials and Methods

### 2.1. Research Model

This research, which examines the relationship between career barrier, presenteeism, and teacher professionalism, was designed in the relational survey model. For this purpose, structural equation modeling (SEM), which is frequently used in relational research, was used as it allows to define predictive relationships between variables and to examine predictive relationships between variables at the same time [42–46]. The hypotheses created within the scope of the research are presented below.

**Hypotheses 1 (H1).** *Family barriers positively affect teacher professionalism.*
**Hypotheses 2 (H2).** *Personal/economic barriers positively affect teacher professionalism.*
**Hypotheses 3 (H3).** *Political/Trade-Union barriers positively affect teacher professionalism.*
**Hypotheses 4 (H4).** *Administrative barriers positively affect teacher professionalism.*
**Hypotheses 5 (H5).** *Higher education barriers positively affect teacher professionalism.*
**Hypotheses 6 (H6).** *Presenteeism positively affects teacher professionalism.*

### 2.2. Study Group

The sample of the study consists of 411 physical education and sports teachers, 169 (41.1%) female and 242 (58.9%) male, who work in Ankara. Convenience sampling method, one of the purposeful sampling methods, was used in the sample selection. A total of 61 (14.8%) of the participants work in primary school, 173 (42.2%) in secondary school and 177 (43.1%) in high school. In addition, 331 (80.5%) of the participants are undergraduate, 73 (17.8%) are graduate and 7 (1.7%) are graduate from a doctoral program. The mean age and standard deviation of the participants was determined as $38.72 \pm 8.68$.

### 2.3. Data Collection Tools

Career Barrier, Presenteeism and Teacher Professionalism Scales as well as the personal information form were used in order to perform the data collection process in the study.

### 2.4. Teachers' Career Barriers Scale

The Teachers' Career Barriers Scale, developed by İnandı & Gılıç (2020), consists of 5 sub-dimensions and 30 items. Sub-dimensions: "family barriers", "personal/economic barriers", "political/trade-union barriers", "administrative barriers" and "higher education barriers". It can be stated that as the score obtained from the scale increases, the career barriers experienced by teachers increase. The Cronbach Alpha coefficient, which shows the internal consistency of the scale, was determined to be 0.93 for the whole scale, 0.93 for the administrative barriers sub-dimension, and 0.86 for the political/trade-union barriers sub-dimension, 0.86 for familial barriers, 0.78 for bureaucratic barriers in higher education, and 0.73 for economic barriers [10]. The Cronbach Alpha coefficient obtained from the data set of the research is 0.91 for the whole scale, and 0.90, 0.85, 0.82, 0.80 and 0.75 for the sub-dimensions, respectively.

### 2.5. The Stanford Presenteeism Scale (SPS-6) (the Problem of Not Being at Work)

The Stanford Presenteeism Scale (SPS-6), which was developed by Koopman et al. (2002) with the contributions of Mark & Co., Inc., Ardmore, PE, USA, and consists of 6 items in total, was used. The low total score obtained from the scale symbolizes a positive situation. The Cronbach Alpha coefficient of the scale is 0.89. The Cronbach Alpha coefficient obtained from the data set was determined as 0.92.

*2.6. Teacher Professionalism Scale*

It was developed by Tschannen-Moran, Parish and Dipaola (2006) and adapted into Turkish by Cerit (2012). The scale, which has a 5-point Likert structure, consists of a total of 8 items and a single dimension. The higher the score obtained from the scale, the higher the level of teacher professionalism. The Cronbach Alpha contribution of the original form of the scale was expressed as 0.90. The Cronbach Alpha coefficient obtained from the data set is 0.87.

*2.7. Confirmatory Factor Analysis Results of Measurement Tools Used in the Scope of the Study*

Confirmatory factor analysis was applied to test the construct validity of Career Barrier, Presenteeism and Teacher Professionalism Scales. The goodness of fit values obtained as a result of the analyses performed are given in Table 1. The obtained fit index values show that the 5-factor structure of the Career barrier Scale and the single-factor structure of the Presenteeism and Teacher Professionalism Scales are confirmed.

**Table 1.** Confirmatory factor analysis results of Career Barrier, Presenteeism and Teacher Professionalism Scale.

| Model Fit Index | Perfect Range | Acceptable Range | CB | P | TP |
|---|---|---|---|---|---|
| $X^2/sd$ | $0 < X^2/sd < 2$ | $2 < X^2/sd < 5$ | 3.31 | 3.62 | 3.46 |
| RMSEA | $0.00 < RMSEA < 0.05$ | $0.05 < RMSEA < 0.10$ | 0.075 | 0.080 | 0.082 |
| PGFI | $0.95 < PGFI < 1.00$ | $0.50 < PGFI < 0.95$ | 0.693 | 0.673 | 0.668 |
| PNFI | $0.95 < PNFI < 1.00$ | $0.50 < PNFI < 0.95$ | 0.706 | 0.726 | 0.695 |
| GFI | $0.90 < GFI < 1.00$ | $0.85 < GFI < 0.90$ | 0.876 | 0.978 | 0.886 |
| AGFI | $0.90 < AGFI < 1.00$ | $0.85 < AGFI < 0.90$ | 0.894 | 0.942 | 0.865 |
| CFI | $0.95 < CFI < 1.00$ | $0.90 < CFI < 0.95$ | 0.932 | 0.990 | 0.927 |

CB: Career Barrier, P: Presenteeism, TP: Teacher Professionalism, [47–50].

*2.8. Data Analysis*

Before analyzing the data, it was checked whether there were missing or erroneous data in the data set. Then, it was examined whether the data set had a normal distribution or not. The Shapiro-Wilk normality test was applied and the data set was found to have a normal distribution by looking at the kurtosis and skewness values [49]. Kaiser-Mayer-Olkin (KMO) coefficient and Bartlett test were used to determine the suitability of the data for factor analysis. According to the results of the analysis, the KMO suitability coefficient was determined as 0.92 for Career Barrier Scale, 0.90 for Presenteeism Scale and 0.89 for Teacher Professionalism Scale. In addition, the Barlett test result was determined to be significant for the measurement tools included in the study ($p < 0.001$). Therefore, these values show the suitability of the data to factor analysis [51]. The theoretically created model was tested with a structural equation model. In the study, descriptive statistics were used in order to determine the mean scores obtained from the scales, and the *t*-test was used to make a comparison with the gender variable. The analyses were carried out using AMOS 22.0, SPSS 22.0 package programs.

## 3. Results

The mean score obtained by the participants was ($M_{ean}$ = 3.29) for the Career Barrier Scale, ($M_{ean}$ = 3.01) for the "family barrier" sub-dimension, ($M_{ean}$ = 3.26) for the "personal/economic barrier" sub-dimension, ($M_{ean}$ = 3.45) for the "political/trade-union barriers" sub-dimension, ($M_{ean}$ = 3.22) for the "administrative barriers" sub-dimension, and ($M_{ean}$ = 3.67) for the "higher education barriers" sub-dimension (Table 2).

The mean score obtained by the participants was determined as ($M_{ean}$ =2.40) for from the Presenteeism Scale and ($M_{ean}$ =3.57) for the Teacher Professionalism Scale (Table 3).

Considering the analysis results, it was determined that there are significant differences between males and females in the career barrier level of the participants in favour of male participants, and in the level of presenteeism and teacher professionalism in a favour of female participants, $t_6(409) = -3.52$, $p < 0.05$; $t_7(409) = -2.16$, $p < 0.05$: $t_8(409) = -2.10$, $p < 0.05$. Aditionally, considering the mean scores of the participants on the sub-dimensions

of the Career Barrier Scale, it was concluded that there was no significant difference between family barriers and higher education barriers considering the participants gender, $t_1(409) = 0.60$, $p > 0.05$; $t_5(409) = -0.11$, $p > 0.05$. A statistically significant difference between males and females was found for the subscales the personal/economic, political/trade-union and administrative barriers in favour of male participants, $t_2(409) = -3.21$, $p < 0.05$; $t_3(409) = -4.21$, $p < 0.05$: $t_4(409) = 3.86$, $p < 0.05$ (Table 4).

**Table 2.** Distribution of the mean scores of the participants from the Career Barrier Scale.

| Scale | N | Min. | Max. | Mean | SD |
|---|---|---|---|---|---|
| Family Barriers | 411 | 1.00 | 5.00 | 3.01 | 0.93 |
| Personal/Economic Barriers | 411 | 1.00 | 5.00 | 3.26 | 0.83 |
| Political/Trade-Union Barriers | 411 | 1.00 | 5.00 | 3.45 | 0.87 |
| Administrative Barriers | 411 | 1.00 | 4.85 | 3.22 | 0.70 |
| Higher Education Barriers | 411 | 1.00 | 5.00 | 3.67 | 0.73 |
| Career Barrier Scale | 411 | 1.63 | 4.63 | 3.29 | 0.56 |

**Table 3.** Distribution of the mean scores of the participants from the Presenteeism and Teacher Professionalism Scales.

| Scale | N | Min. | Max. | Mean | SD |
|---|---|---|---|---|---|
| Presenteeism Scale | 411 | 1.00 | 4.67 | 2.40 | 0.89 |
| Teacher Professionalism Scale | 411 | 1.88 | 5.00 | 3.57 | 0.66 |

**Table 4.** *t*-test results of the mean scores of the participants from the Career Barrier, Presenteeism and Teacher Professionalism Scales according to the gender variable.

| Variable | Gender | N | Mean | SD | Df | t | p |
|---|---|---|---|---|---|---|---|
| FB | Female | 169 | 3.04 | 0.97 | 409 | 0.60 | 0.559 |
| | Male | 242 | 2.98 | 0.92 | | | |
| PEB | Female | 169 | 3.11 | 0.76 | 409 | −3.21 | **0.001** |
| | Male | 242 | 3.37 | 0.86 | | | |
| PTB | Female | 169 | 3.24 | 0.85 | 409 | −4.21 | **0.000** |
| | Male | 242 | 3.60 | 0.86 | | | |
| AB | Female | 169 | 3.06 | 0.70 | 409 | −3.86 | **0.000** |
| | Male | 242 | 3.33 | 0.69 | | | |
| HEB | Female | 169 | 3.67 | 0.74 | 409 | −0.11 | 0.907 |
| | Male | 242 | 3.68 | 0.73 | | | |
| CB | Female | 169 | 3.17 | 0.54 | 409 | −3.52 | **0.000** |
| | Male | 242 | 3.37 | 0.56 | | | |
| P | Female | 169 | 2.28 | 0.88 | 409 | −2.16 | **0.032** |
| | Male | 242 | 2.47 | 0.89 | | | |
| TP | Female | 169 | 3.64 | 0.68 | 409 | 2.10 | **0.046** |
| | Male | 242 | 3.50 | 0.66 | | | |

FB: Family Barriers, PEB: Personal/Economic Barriers, PTB: Political/Trade-Union Barriers, AB: Administrative Barriers, HEB: Higher Education Barriers, CB: Career Barrier, P: Presenteeism, TP: Teacher Professionalism.

When Table 5 is examined, it is seen that there is a low-level relationship between the family, personal/economic, political/trade-union, administrative and higher education barriers of the participants and presenteeism and teacher professionalism, ($r_1 = 0.16$, $-0.14$, $p < 0.01$, $r_2 = 0.17$, $-0.18$, $p < 0.01$, $r_3 = 0.16$, $-0.14$, $p < 0.01$, $r_4 = 0.28$, $-0.28$, $p < 0.01$, $r_5 = 0.12$, $0.19$, $p < 0.01$. With the determination of the relationship between the variables in the study, the structural equation model was tested (Table 5).

**Table 5.** Examination of the relationship between variables with the Pearson Product Moment Correlation.

| Variable | FB | PEB | PTB | AB | HEB | P | TP |
|---|---|---|---|---|---|---|---|
| FB | 1 | | | | | | |
| PEB | 0.45 ** | 1 | | | | | |
| PTB | 0.21 ** | 0.35 ** | 1 | | | | |
| AB | 0.19 ** | 0.27 ** | 0.56 ** | 1 | | | |
| HEB | 0.18 ** | 0.33 ** | 0.35 ** | 0.44 ** | 1 | | |
| P | 0.16 ** | 0.17 ** | 0.16 ** | 0.27 ** | 0.12 ** | 1 | |
| TP | −0.14 ** | −0.18 ** | −0.14 ** | −0.28 ** | 0.19 ** | −0.14 ** | 1 |

** $p < 0.01$, FB: Family Barriers, PEB: Personal/Economic Barriers, PTB: Political/Trade-Union Barriers, AB: Administrative Barriers, HEB: Higher Education Barriers, P: Presenteeism, TP: Teacher Professionalism.

The fit index values seen in Figure 1 are given in Table 6.

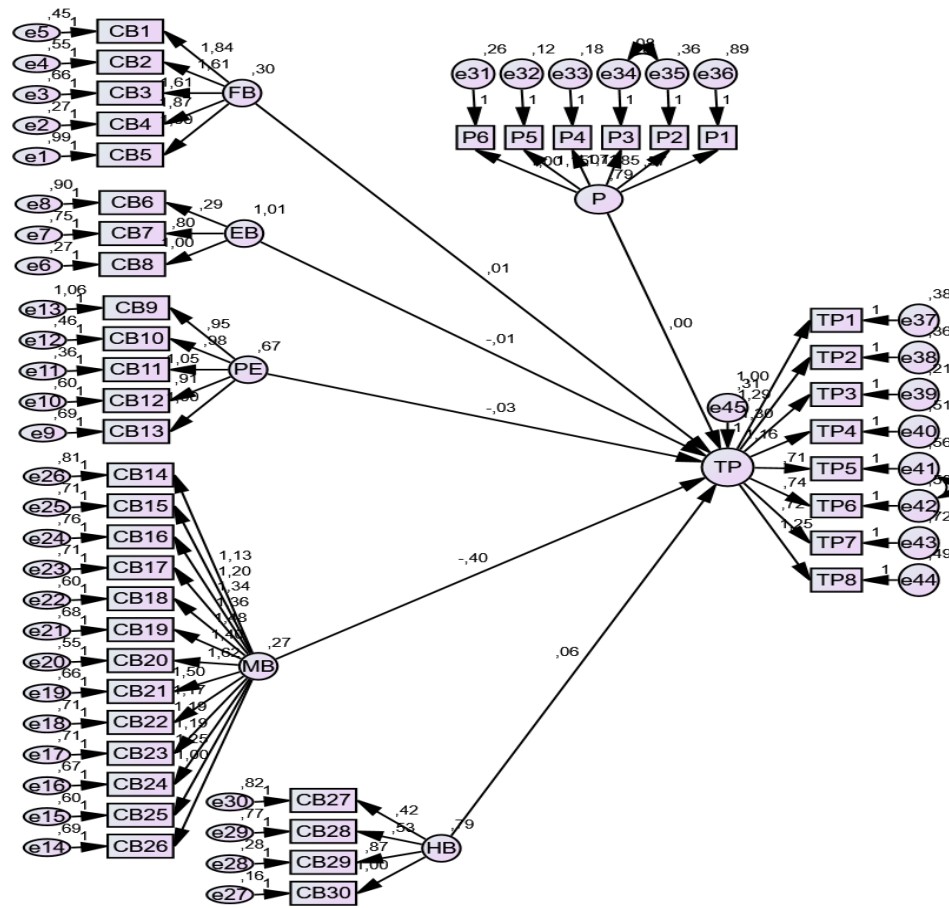

**Figure 1.** Structural equation model.

**Table 6.** Structural equation model fit index values.

| Model Fit Index | Perfect Range | Acceptable Range | SEM |
|---|---|---|---|
| $X^2/sd$ | $0 < X^2/sd < 2$ | $2 < X^2/sd < 5$ | 2.91 |
| RMSEA | $0.00 < RMSEA < 0.05$ | $0.05 < RMSEA < 0.10$ | 0.06 |
| PGFI | $0.95 < PGFI < 1.00$ | $0.50 < PGFI < 0.95$ | 0.79 |
| PNFI | $0.95 < PNFI < 1.00$ | $0.50 < PNFI < 0.95$ | 0.81 |
| GFI | $0.90 < GFI < 1.00$ | $0.85 < GFI < 0.90$ | 0.86 |
| AGFI | $0.90 < AGFI < 1.00$ | $0.85 < AGFI < 0.90$ | 0.83 |
| CFI | $0.95 < CFI < 1.00$ | $0.90 < CFI < 0.95$ | 0.91 |

SEM: Structural Equation Model, [47–50].

Considering Table 5, it can be stated that the model meets the necessary goodness of fit criteria, ($x^2$/sd = 2.91, RMSEA = 0.06, PGFI = 0.79, PNFI = 0.81, GFI = 0.86, AGFI = 0.83, CFI = 0.91). After examining the goodness of fit index values of the model, the paths in the model and the parameter estimates for the model were examined. According to the model, standardized β coefficients, standard error, critical ratio, *p* and R2 values among variables are given in Table 7.

**Table 7.** Structural equation model results.

| Variables | | Standardized β | Standard Error | Critical Ratio | *p* | $R^2$ |
|---|---|---|---|---|---|---|
| Family B. | | 0.013 | 0.06 | 0.25 | 0.801 | |
| Personal/Economic B. | | −0.015 | 0.03 | −0.27 | 0.787 | |
| Political/Trade-Union B. | Teacher | −0.047 | 0.04 | 0.87 | 0.385 | |
| Administrative B. | Professionalism | −0.343 | 0.07 | −5.52 | *** | 0.13 |
| Higher Education B. | | 0.093 | 0.04 | 1.75 | 0.079 | |
| Presenteeism | | −0.005 | 0.03 | −0.09 | 0.928 | |

*** highly significant.

When the analysis results are examined, it is concluded that administrative barriers affect teacher professionalism ($\beta_4$ = 0.343; $p < 0.05$). Furthermore, it was determined that family, personal/economic, political/trade-union, higher education barriers and presenteeism did not affect teacher professionalism ($\beta_1$ = 0.013; $p > 0.05$; $\beta_2$ = −.015; $p > 0.05$; $\beta_3$ = −.047; $p > 0.05$; $\beta_5$ = 0.093; $p > 0.05$; $\beta_6$ = −.005; $p > 0.05$). According to these findings, the hypotheses numbered 1, 2, 3, 5 and 6 of the research were rejected, while the hypothesis numbered 4 was accepted. Considering the Squared Multiple Correlations (R2) value of the model, it was determined that family, personal/economic, political/trade-union, administrative, higher education barriers and presenteeism explained 13% of teacher professionalism.

## 4. Discussion and Conclusions

Within the scope of the study, it was aimed to determine the effect of career barrier and presenteeism level of physical education and sports teachers on teacher professionalism. In addition, it was determined whether the results of descriptive statistics and the mean scores obtained from the scales in the study caused a difference according to the gender variable.

Considering the mean scores of the participants from the Career Barrier Scale, it can be stated that the difficulties they experienced are above average and the barriers originating from higher education have a significant point average (Table 2). It is seen that there are studies, which support the research finding, in the literature [52–54]. It can be stated that no possibility of relocation for postgraduate education, the lack of flexibility for postgraduate education due to working hours, and the inability to provide a separate postgraduate quota are effective at this point.

Considering the mean score of the participants from the Presenteeism Scale, it can be stated that the level of presenteeism is low and positive (Table 3). From this point forward, it can be said that the participants are able to cope with the stressful situations they experience in the school environment, that they do not experience any physical and mental problems in the school environment, and that they generally have the energy to fulfill their duties. However, it is observed that the professionalism level of the participants is not at a high level (Table 3). Similar results are seen when the relevant literature is considered [41,46,55,56]. It can be said that providing social support to colleagues, doing their job willingly, helping students, and developing a cooperative behavior will contribute to participants. It can be stated that this may cause problems when the desire for a qualified education is taken into consideration.

When the career barrier level of the participants was compared according to the gender variable, it was concluded that male participants had more career barriers (Table 4). This may be due to the fact that male teachers have more desire to support their families and

advance in the business life. There are studies in the literature that differ from the findings of the research [57–59]. The reason for this is assumed to be due to the cultural differences in the study groups. In addition, it was determined that male participants experienced presenteeism problem more than female participants. This result shows that the focus problem experienced by male participants in the work environment is more noticeable than that of female participants. It is assumed that the reason for this result is that women feel a sense of responsibility. Moreover, it was concluded that the professionalism level of female participants was significantly higher than that of male participants. Considering the teaching profession, it is predicted that the level of professionalism is high due to the higher professional satisfaction level and responsibility acquisition of women. When the literature is reviewed, it is seen that related studies support this result [60–62].

Considering career barriers, it was concluded in the research that administrative barriers affect teacher professionalism, while presenteeism behaviors do not affect the level of professionalism (Table 7). Although the research was carried out during the Covid-19 Pandemic period, it can be interpreted that both the presenteeism behaviors and professionalism levels of the teachers in our target group are not under the influence of the natural negatives that this process may bring along. Administrators" difficulties due to their personal whims and thoughts in participating in organizations focused on personal development, ignoring the efforts of teachers, insufficient transfer of encouraging practices related to career assessments, and the inability or willingness of university administrations to contribute to the career development of teachers has emerged as a barrier to the increase in teachers' professionalism level.

School administrators have responsibilities in creating a quality school climate [63,64]. Akbaşlı and Diş (2019) emphasized that school administrators should be assertive and encouraging for the development of teachers. Additionally, Bateman and Snell (2016) stated that it is important for administrators to take the guiding role by listening to teachers' ideas. Therefore, it can be said that the attitudes and behaviors of school administrators have an important role in the development of teachers, and that a collaborative approach in career planning will result in a development-oriented result. From this point of view, in order to increase the level of professionalism of physical education and sports teachers, it is recommended to increase the level of awareness of school administrators on collaborative management, career development and effective communication through in-service training. It is thought that the development of school administrators on this subject will contribute to the personal development of teachers. Furdermore, reducing the effectiveness of career barriers for teachers is considered important. Structural barriers in particular need to be carefully investigated. It is possible to prevent teachers from living with career disabilities through practices such as entitlement to postgraduate education with additional quotas, regulation of inefficient school management, making teaching a career profession like being an academician, and increasing salary opportunities. In addition, in order to examine the factors affecting professionalism in depth, it is recommended to carry out similar studies using qualitative research methods and to observe the characteristics of teachers, who are considered to be one of the most important subjects of quality in education, through longitudinal research series.

**Author Contributions:** Conceptualization, N.B.G. and S.K.; methodology, N.B.G.; software, N.B.G.; validation, N.B.G. and L.-G.T.; formal analysis, N.B.G.; investigation, N.B.G. and E.L.İ.; resources, S.K. and F.S.; data curation, E.L.İ. and F.S.; writing—original draft preparation, N.B.G.; writing—review and editing, N.B.G., S.K., E.L.İ. and F.S.; visualization, F.S., E.L.İ. and L.-G.T.; supervision, F.S., E.L.İ. and L.-G.T.; project administration, N.B.G. All authors have read and agreed to the published version of the manuscript.

**Funding:** This research received no external funding.

**Institutional Review Board Statement:** Ethical approval was obtained from Balıkesir University Faculty of Sport Sciences for the research.

**Informed Consent Statement:** Informed consent was obtained from all subjects involved in the study.

**Data Availability Statement:** The data set will be sent to the researchers by the responsible author.

**Conflicts of Interest:** The authors declare no conflict of interest.

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
