# Peer review of "Examination of the Effect of Career Barriers and Presenteeism Behaviors on Teacher Professionalism through Structural Equation Model"

_education, doi:10.3390/educsci12120933_

Round 1
Reviewer 1 Report
In the Article 2. Methods and Techniques there is no indication which technique have been studied on teachers of sports. Was CATI, CAPI, CAWI or PAPI used? I propose to draw some practical conclusions in the section entitled ’Discussion and conclusions' (4). How can professional barriers to the professionalism of certified teachers be removed?
Author Response
Thank you for your comments. They help us improve our work.
We made changes in the article in order to resolve the observations made.
Please see the attachment

Reviewer 2 Report
Dear Sir/ Madam,
The topic of the paper is very interesting and it was my pleasure to read it.
The title of the paper is adequate to the problem of the study. The methodology of the research fits the studied subject. Professional terminology is properly used. As a reviewer of this paper, I find that there are no errors in theoretical presentation, but there are some suggestions to correct in the paper:
1. Each table should have legend explaining the abbreviations used in the table,
2. For the same word use always the same abbreviation
3. Abbreviation for standard deviation is „SD“ (table 2, 3 and 4),
4. Instead of the abbreviation for Arithmetic Mean use „Mean“ to be more pronounced when interpreting the table
5. In the table 4 I suggest that the statistically significant values are marked with “*” in order to be more pronounced when interpreting the table, also abbreviation for „degree of freedom is „df“ not „sd“,
6. In the text when author is interpreting the results Table number should be given in the brackets. (Example: „The mean score obtained by the participants (Table 2) was…“)
7. Page number 6 paragraph 2 change into:
„Considering the analysis results, it was determined that there are significant differences between males and females in the career barrier level of the participants in favour of male participants, and in the level of presenteeism and teacher professionalism in a favour of female participants, t6(409) = −3.52, p < .05; t7(409) = −2.16, p < .05: t8(409) = −2.10, p < .05. Also, considering the mean scores of the participants on the sub-dimensions of the Career Barrier Scale, it was concluded that there was no significant difference between family barriers and higher education barriers considering the participants gender, t1(409) = .60, p > .05; t5(409) = −.11, p > .05. A statistically significant difference between males and females was found for the subscales the personal/economic, political/trade-union and administrative barriers in favour of male participants, t2(409) = −3.21, p < .05; t3(409) = −4.21, p < .05: t4(409) = 3.86, p < .05.“
Parts of the paper are of the appropriate extent and there are no unnecessary repetitions in the text. The text is written clearly and logically, and the conclusion is drowning from the results obtained. As for the literature, references are written correctly.
Author Response

(The authors gave the same response as above.)
